# Electrospun Antimicrobial Drug Delivery Systems and Hydrogels Used for Wound Dressings

**DOI:** 10.3390/pharmaceutics16010093

**Published:** 2024-01-10

**Authors:** Zahra Moazzami Goudarzi, Angelika Zaszczyńska, Tomasz Kowalczyk, Paweł Sajkiewicz

**Affiliations:** Laboratory of Polymers and Biomaterials, Institute of Fundamental Technological Research, Polish Academy of Sciences, Pawińskiego 5B, 02-106 Warsaw, Poland; zmoazami@ippt.pan.pl (Z.M.G.); azasz@ippt.pan.pl (A.Z.); psajk@ippt.pan.pl (P.S.)

**Keywords:** wound dressings, drug delivery systems, thermoresponsive hydrogels

## Abstract

Wounds and chronic wounds can be caused by bacterial infections and lead to discomfort in patients. To solve this problem, scientists are working to create modern wound dressings with antibacterial additives, mainly because traditional materials cannot meet the general requirements for complex wounds and cannot promote wound healing. This demand is met by material engineering, through which we can create electrospun wound dressings. Electrospun wound dressings, as well as those based on hydrogels with incorporated antibacterial compounds, can meet these requirements. This manuscript reviews recent materials used as wound dressings, discussing their formation, application, and functionalization. The focus is on presenting dressings based on electrospun materials and hydrogels. In contrast, recent advancements in wound care have highlighted the potential of thermoresponsive hydrogels as dynamic and antibacterial wound dressings. These hydrogels contain adaptable polymers that offer targeted drug delivery and show promise in managing various wound types while addressing bacterial infections. In this way, the article is intended to serve as a compendium of knowledge for researchers, medical practitioners, and biomaterials engineers, providing up-to-date information on the state of the art, possibilities of innovative solutions, and potential challenges in the area of materials used in dressings.

## 1. Introduction

The human body includes an organ known as the skin, which is the largest in size [1]. On average, an adult’s skin covers about 2 square meters of surface and can vary in thickness from 1 to 4 mm. The thickest skin is found on the hands and feet, and in individuals working all their lives physically, it can be even 10 mm thick. The skin’s primary function is to cover the internal organism, thus providing a barrier from the external environment. However, diabetes, cuts, and illnesses can influence the structure and function of this organ [2,3]. The skin is composed of two layers: the dermis and the epidermis. The epidermis, an external layer made of connective tissue, contains numerous capillaries. Its primary function is to act as a mechanical and chemical barrier against external factors (heat, cold, UV radiation), mechanical injuries, and chemical substances, and the penetration of microorganisms also protects the internal organs and regulates the water–electrolyte, vitamin, and fat balance in the body [4]. Over the last few decades, there have been enormous developments in materials dedicated to medicine, particularly nanosized materials. Electrospun nanosized materials produced via electrospinning are investigated and applied in various biomedical fields, such as muscle and neural tissue [5,6,7], urology [8,9], drug delivery systems [10,11,12], regeneration [13,14], cartilage [15], anti-cancer treatment [16], and skin, especially wound dressings (Figure 1A) [17,18,19].

Nanomaterials have unique features, such as a high surface-to-volume ratio, and innovative properties, such as hydrophobicity, morphology, or surface charge, that can be easily modified [20,21,22]. Thus, it can mimic the extracellular collagen matrix (ECM) and simultaneously avoid the clearance properties of the human immune system [23,24]. According to recent investigations [25], cells attach better to the smaller diameters of the fibers. Abrigo et al. reviewed wound dressings and gave an insight into the evolution of wound dressing classification and application [26]. Currently, there are 50 million patients suffering from severe wounds. Moreover, in the US, USD 25 billion is spent annually on chronic wound treatment per year [27]. In the UK, this is GBP 4.5–GBP 5.1 billion [28], and in the Scandinavian countries, around 2–4% of the total healthcare budget [29]. This economic problem requires widespread scientific investigation. The global wound care market is expected to surpass USD 2.2 billion by 2024 [30].

Wound dressings serve as a crucial protective measure against infections, injuries, and the absorption of wound secretions. In an effort to improve wound-healing outcomes and prevent infections, researchers are actively exploring various methods of developing electrospun dressings. The process of wound healing is a complex and intricate procedure that occurs within the human body [31].

Within this manuscript, we meticulously examine the advancements in wound dressing research and provide a comprehensive overview of different types of wound dressings and their potential uses. Our focus is on presenting the latest developments in the creation of electrospun nanofibers as wound dressings, emphasizing innovative strategies for constructing state-of-the-art systems. Furthermore, we explore the future prospects and challenges that lie ahead in this field.

## 2. Commercial Wound Dressings

Currently, various types of wound dressings are being researched for wound-healing management. Commercial dressings can be divided into two main categories: traditional dressings and more complex structures that include healing agents [32]. Additionally, currently, the available wound dressings can be summarized in detail as interactive, passive, bioactive, and advanced (Figure 1B) [33].

The most popular market dressings are available as foam [34], film [35], sponge [36,37], and hydrogel [38,39]; nanofibrous membranes are mainly in a stage of non-clinical testing [40]. Although over 3000 types of dressings are on the market, there is no ideal dressing for every type of wound. This is the main reason for the further intensive work of scientists worldwide. The most important issue is to fit the dressings to the specified types of wound. Every product has many advantages but also disadvantages (Table 1).

There are many different types of wound dressings available, but there has not been enough research conducted to compare their characteristics. Scientists are working on developing new and advanced types of dressings, like smart bandages made from polymers [48]. Many researchers strive to design dressings with superior qualities compared to those already on the market. However, the state-of-the-art findings are seldom compared to the performance of commercial dressings. 

### 2.1. Traditional Wound Dressings

Conventional wound-dressing products encompass lint, plasters, gauze, bandages (both natural and synthetic), and cotton wool, which are used as primary or secondary dressings to safeguard wounds against contamination. Traditional wound dressings provide a safeguard against external infections for the wound. Their ability to effectively regulate moisture absorption from the wound is limited, resulting in insufficient dampness for prompt healing. These wound dressings are commonly employed for wounds with minor discharge or as backup dressings [49]. 

Bandages of natural cotton wool, cellulose, or synthetic materials like polyamide can secure lightweight dressings. An example of a natural bandage can be Xeroform™, which is suitable for non-exudating to mildly exudating wounds. Another type, gauze dressings, consisting of woven and nonwoven fibers like cotton, rayon, and polyester, offer a certain level of protection against bacterial infections. Gauzes are cheap and can be used as wound covers. Additionally, gauze dressings tend to become moist and adhere to the wound, causing discomfort and secondary damage during removal due to excessive adhesion. Tulle dressings, such as Bactigras, Jelonet, and Paratulle, are commercially available impregnated dressings containing paraffin and are appropriate for superficial clean wounds. To summarize, conventional wound dressings are dedicated to dry and clean wounds [50].

### 2.2. Interactive and Bioactive Wound Dressings

The group of interactive and bioactive wound dressings consists of hydrogels, semi-permeable films, foam dressings, hydrocolloids, and hydrofibres.

Hydrogels are hydrophilic, biocompatible, and permeable to nutrients and metabolites, making them non-irritating materials made from synthetic polymers [51]. Scientists [52] have observed that hydrogel dressings are suitable for all four stages of wound healing except for infected and heavily draining wounds. Numerous studies have shown that hydrogel dressings are effective in treating chronic leg ulcers. However, their low mechanical strength and potential to cause skin infections are major drawbacks [53].

Further, another type of commonly used wound dressing comprises semi-permeable films. Semi-permeable films are dedicated to treating non-exudative or low-exudative wounds [54]. Also, they have unique properties such as transparency, elasticity, and flexibility [55]. It is worth mentioning other dressings, such as Opsite, Tegader, Mefilm, and Biooclusive. They are recommended for superficial and epithelializing wounds [56].

Subsequently, semi-permeable foam dressings consist of hydrophobic and hydrophilic foam materials, sometimes with adhesive borders. Typically, foam dressings serve as primary dressings due to their high absorbency and moisture vapor permeability, eliminating the need for additional secondary dressings [57]. Foam dressings can absorb varying amounts of wound drainage, depending on the thickness of the wound. Both adhesive and non-adhesive foam dressings are available. Foam dressings are suited to exudating wounds and also granulating wounds [58].

Bioactive wound dressings, more precisely, hydrocolloids and hydrofibres, can promote healing and offer biocompatibility, biodegradability, and non-toxicity properties; however, this type of dressing cannot conduct water vapor exchange, which is a drawback to application to infected wounds that need oxygen for improved healing. They can be derived from various sources, including natural tissues and artificial materials [59]. Examples of such materials include collagen [60], hyaluronic acid (HA) [61], chitosan [62], alginate, elastin, dextran, and gelatin [63]. The use of bioactive dressings with growth factors and antimicrobial agents enhances wound healing. Collagen, a major structural protein, plays a significant role in natural healing [64], as does HA [65]. Collagen promotes the formation of fibroblasts, while HA is naturally biocompatible, biodegradable, and non-immunogenic [66,67]. Studies have shown that nanofibrous wound dressings made of a polyurethane–HA blend with integrated propolis are suitable for wound dressing [68]. There are many commercially available wound dressings that contain HA, such as Hyalomatrix or Hyalosafe. Hyalomatrix is a flexible, adaptable bilayered dermal material that promotes the regeneration of the dermis and the closure of wounds. Hyalosafe is a transparent film utilized in treating wounds, especially second-degree burns [69].

## 3. Electrospinning as a Method for Wound Dressing Formation

Electrospinning is an advanced technique for producing nanofibers using electricity. This process enables the production of nanofibers with a diameter from several nanometers to several micrometers. Electrospinning is used in many fields of medicine, such as biomedicine, material engineering, and electrical engineering (Figure 2) [70]. Nanofibers obtained via electrospinning usually have a very small diameter, which is advantageous in various applications, especially in dressings. Additionally, the structure of nanofibers generated via electrospinning is characterized by a large surface area in relation to the volume, which is important in the context of the adsorption, transportation of substances, and cellular interactions that can take place in dressings [71]. High porosity may be beneficial in the processes of wound healing, fluid absorption, and interaction with cells. As well as regulating the structure and morphology, by controlling the electrospinning parameters, the structure, morphology and mechanical properties of nanofibers can be adjusted, adapting them to specific application needs [72]. Nanofiber-based products play a key role in wound healing due to their unique properties that contribute to the creation of a favorable microenvironment for the wound-healing process. The exceptionally high surface-to-volume ratio increases the adsorption and retention of bioactive molecules, growth factors, and other therapeutic agents on the surface of the nanofibers, promoting their controlled release to the wound site. Additionally, nanofibers can support the formation of new blood vessels (angiogenesis) [73].

Nanofiber-based dressing products can result in improved blood flow to the wound site and improve the delivery of nutrients and oxygen, facilitating cell proliferation, migration, and overall damaged-skin regeneration [74].

Various manufacturing techniques have been devised to produce porous architectures that have antibacterial properties, such as phase separation, supercritical fluid, hydrogel, and electrospinning (Figure 3) [75,76,77,78,79]. Although electrostatic spinning has become popular in the field of biomedical engineering in recent years, its history can be traced back to the 1600s, when English physicist William Gilbert first described the electrostatic phenomena [80]. However, electrospinning has garnered significant attention due to its simplicity, cost effectiveness, versatility, and scalability, and the ability to control fiber morphology during the process [81,82,83]. Electrospraying produces micro- and nanoparticulate materials smaller than typical spray-dried particles, resulting in a high surface-to-volume ratio [80].

The electrospinning apparatus is gaining momentum in commercialization. Among the widely adopted electrospinning techniques, melt electrospinning [86], wet electrospinning [87], coaxial electrospinning [85], and self-bundling electrospinning [88] are the most commonly utilized. Electrospinning can generate polymer nanofibers (with diameters ranging between 50 and 1000 nm) through techniques such as wet or hot melt electrospinning [86,87].

Electrospinning has the potential to create a large surface area and porosity, facilitating cell attachment and simplifying the exchange of nutrients and waste products within the anatomical sites of implantation [77,81]. Compared to conventional systems, electrospun materials exhibit a superior performance in the wound-healing process, owing to their unique morphology and structure that resemble those of the ECM, high specific surface area, good draining capacity, nutrient and metabolite exchange, and air permeability. Moreover, drugs, antimicrobials, and bioactive molecules can be readily incorporated into, or attached to, the surface of electrospun mats to promote regeneration. Notably, it has been observed that compounds offering antioxidant activity can facilitate the wound-healing process [12,89,90].

Electrospinning is a highly flexible and affordable process, but its execution can be intricate due to various factors that impact the ultimate structure and properties of the electrospun fibers. The key factors among these are the parameters of the solutions employed, including the viscosity, the polymer concentration, the molecular weight of the polymers, and the conductivity. The process of electrospinning can be influenced by several factors, including the fabrication process and environmental conditions such as the voltage, the distance between the collector and tip, the flow rate, the humidity, and the temperature [77,79,81,91]. By adjusting the independent variables of the polymer concentration, conductivity, flow rate, and voltage settings, the diameter and dimensions of fibers can be manipulated [77,81,91]. Furthermore, empirical evidence suggests that incorporating electrospun fibers into fabric-reinforced composites can significantly enhance their mechanical properties. Electrospinning yields materials with a high surface area and porosity, facilitating cellular adhesion and nutrient and waste exchange at implantation sites [77,81].

The chemical and physical properties of electrospun nanofibers, including their composition, degradation, diameter, strength, porosity, and incorporated bioactive molecules, can influence their interaction with injured tissue and its biological environment. Consequently, these features directly affect the effectiveness and performance of dressings made from electrospun nanofibers. Additionally, the architecture and structure of the dressings significantly impact the wound-healing process, which involves a cascade of events, including hemostasis, inflammation, proliferation, and tissue remodeling. Growth factors, which are biologically active polypeptides, play a crucial role in controlling cell growth, proliferation, and migration during the wound-healing process and can regulate all stages. Moreover, the role of vitamins in the wound-healing process is also noteworthy [92,93,94].

Developing advanced materials for wound dressings is a challenging and yet-to-be-solved task. These materials must act as temporary skin substitutes, serving multiple purposes, including absorbing fluids, preventing infections, and assisting cell growth and movement to support the skin’s healing process [89]. Trauma, surgery, and pathological diseases can cause skin wounds that compromise the skin’s integrity, often resulting in permanent physical defects and even death, causing a major healthcare concern. Treating wounds has become a significant medical problem worldwide due to the aging population and an increasing prevalence of metabolic and cardiovascular diseases, leading to a higher incidence of chronic wounds. Wound dressings are crucial to the healing process, leading to scientific research in developing bioactive materials that mimic the ECM, promoting cell adhesion and migration to the wound site to facilitate skin regeneration (Figure 4) [95]. Nanostructured materials offer promising solutions to address major issues associated with skin regeneration, such as scar formation, poor tissue integration, and bacterial infection. Bacterial contamination is particularly critical in wound care since it can impede healing and lead to chronic wounds. The ideal wound dressing material should be an elastic, biocompatible, and biodegradable system with antimicrobial activity, capable of absorbing wound fluids and exudates, regulating nutrient and gas exchange, maintaining a moist local environment, supporting cell proliferation and migration to aid the healing process, reducing patient discomfort, and avoiding scarring [89,92,93,94,96].

A review of the literature reveals successful in vivo studies proving the use of various types of material for the regeneration of damaged skin. Scientists [97] tested citrus pectin stabilizing ZnO nanoparticles on 8-week-old mice. ZnO nanoparticles were evenly coated on the surface of the Coll/CS fibrous structure. At different analysis periods, the degree of wound closure was measured on days 3, 7, 14, and 21. Increased antimicrobial inhibition against *Staphylococcus aureus* and *Escherichia coli* was observed. There was also an improvement in the activity regarding extracellular matrix formation on newly formed tissues. Such nanoparticles embedded in fibers can be an effective wound dressing for the treatment of burns. Another study conducted on rats [98], using a nanofibrous material based on polycaprolactone (PCL), polyvinyl alcohol (PVA), poly (vinylidene fluoride-co-hexafluoropropene) (PVdF-HFP), polyacrylonitrile (PAN), and polymer blend of polyurethane (PEU) and PAN, has proven the beneficial effect of nanofibers on wound healing. Additionally, added silver nanoparticles increased the antibacterial properties. Further in vivo studies were performed on diabetic rats. Electrospun nanofibers based on polycaprolactone (PCL) and gum tragacanth (GT) with the addition of curcumin showed rapid wound closure with well-formed granulation tissue. Wounds treated with seeded scaffolds showed a lesser scab area, and the histochemical results showed a significantly improved healing efficiency with scaffold stem cells [99].

## 4. Antibacterial Nanofibers for Wound Dressing

In particular, chronic infections often stem from bacterial sources, proliferating rapidly within pre-existing wounds. This underscores the critical necessity of employing antibacterial substances. Due to their substantial surface area, antibacterial nanofibers enable the effective incorporation of antibacterial agents [100]. Antibacterial wound dressings are crucial to preventing infections that impede healing [26]. Infections can be attributed to microorganisms originating from hospitalized patients, internal sources, or the surrounding skin [101]. Therefore, it is crucial to treat wounds with dressings that incorporate antibiotics and antibacterial substances to prevent bacterial infections. As a result, alternative antibacterial agents, such as quaternary ammonium compounds, metal ions, nanoparticles, and antimicrobial polymers, have been suggested [102].

Wound healing can be divided into three stages, i.e., the closing of damaged blood vessels and wound edges, then wound cleansing and, finally, the regeneration and reconstruction of damaged tissues, i.e., the formation of a new epidermis or scar [103]. Wound healing is an extremely complicated process, so developing the perfect material for a dressing is an extraordinary challenge for scientists. One way is to develop nanofiber-based materials to accelerate wound healing using electrospinning. The main advantage of electrospinning is controlling the size of the nanofibers, which allows the obtention of very fine structures [104]. The small size of nanofibers has the potential to increase the specific surface area, which can improve the solubility of drugs and increase their effectiveness [105]. The resulting homogeneous structure of nanofibers can also improve the stability of drugs, and the extensive surface area of nanofibers promotes drug adsorption. This, in turn, may lead to an increased bioavailability of drugs, i.e., their easier absorption by the body. Electrospun materials are characterized by the possibility of controlled release. Using electrospinning, drug carriers can be constructed from nanofibers with specific properties, which enable controlled drug release. This process allows us to adjust the rate and manner in which the drug is released, which may be important for drugs with a specific pharmacokinetic profile [106].

The adhesion of bacteria to a wound’s surface results in biofilms, densely populated areas containing bacteria. These biofilms can potentially protect bacteria from the immune system and antibiotics. Consequently, releasing endotoxins can lead to sepsis and, ultimately, death [107]. Traditionally, antibiotics like penicillin and methicillin have been used in wound dressings and on the skin surrounding the wound [108]. The use of conventional antibiotics is limited by the emergence of antibiotic-resistant bacteria. Antibacterial materials often have inadequate efficacy or induce cytotoxic effects.

Antimicrobial peptides (AMPs) have garnered significant attention as a novel class of antimicrobial agents. Naturally occurring in a range of organisms, including mammals, fish, insects, amphibians, and even some bacteria, these antibiotics play a crucial role in bolstering the host’s immune system’s defenses against bacteria, fungi, and viruses [109]. Despite their diverse structures, AMPs typically contain cationic and amphiphilic regions with an α-helical conformation, which can potentially damage bacterial cell membranes [110]. AMPs have demonstrated rapid and remarkable efficacy in killing many bacteria [111]. Moreover, AMPs have been shown to promote wound healing by enhancing re-epithelialization and angiogenesis, neutralizing lipopolysaccharides (LPS), and modulating the immune response [112].

Some biopolymers, such as chitosan, have natural antibacterial properties [113]. However, other types of polymeric wound dressings require the addition of antibacterial agents to prevent infections. In nanofibrous wound dressings, there are various categories of antibacterial agents that have been researched and developed, depending on the specific agent and how it is incorporated into the nanofibers. In the realm of biomedical applications, both synthetic and natural biopolymers have strengths and weaknesses [12,77,81,82]. Nanomaterials are emerging as promising antibacterial agents, with the research predominantly focused on developing advanced wound dressings with both hemostatic and antibacterial attributes to meet contemporary societal demands [114].

Effective wound dressings are pivotal in infection prevention, but their use potentially exacerbates bacterial resistance (Figure 3) [37,115]. Researchers address this issue by integrating antibacterial agents into materials intrinsically endowed with antibacterial properties. These materials encompass various polymers, including chitosan, collagen, alginate, and silver (Ag), copper (Cu), zinc oxide (ZnO), and gold (Au) nanoparticles (Figure 5), as documented in Table 2 and Table 3 [76,85,92,96,116,117,118]. Dressings endowed with intrinsic antibacterial efficacy offer sustained benefits and reduced cytotoxicity compared to those releasing antibacterial agents, as shown in Figure 3 [84,85].

Chitosan, renowned for its innate antibacterial properties, exhibits promise in hydrogel wound dressings. Nevertheless, its antibacterial effectiveness decreases in non-acidic environments, posing challenges to the formulation of in situ antibacterial hydrogel dressings [79,120,121].

### 4.1. Biopolymeric Nanofibrous Wound Dressings Containing Antibacterial Nanoparticles

Silver nanoparticles (Ag NPs) are among the most studied metal nanoparticles due to their unique properties and demonstrated antibacterial effect. Recently, researchers have developed a biomimetic electrospun nanofibrous wound dressing that incorporates both chitosan and Ag NPs to address bacterial infection and promote tissue repair [89]. Altangerel et al. [76] have developed a simple way to create nanofibrous scaffolds in 3D that have strong antibacterial properties, which are particularly useful in treating slowly healing diabetic wounds. Octenidine (OCT) is a cationic surfactant and antimicrobial agent that can transform a 2D membrane into a hydrophilic 3D scaffold in a “two birds, one stone” manner. By combining OCT with Ag NPs, the 3D multilayered porous scaffold has even greater antibacterial effectiveness than the 2D membrane. Additionally, the results of in vitro tests on mouse fibroblasts, L929, have confirmed that this 3D scaffold is not toxic to cells. These findings suggest that this multifunctional 3D scaffold could be an excellent solution for treating diabetic wounds and repairing skin.

Infection and resistance in wounds are health concerns that may be reduced with antibacterial wound dressings. Healing various skin wounds is a lengthy process often combined with bacterial infection and scar formation. A biomimetic electrospun nanofibrous wound dressing loaded with materials with dual antibacterial and tissue repair properties could be developed to address this problem [78,96,119,122,123,124].

**Table 2 pharmaceutics-16-00093-t002:** Antibacterial electrospun fibers for wound dressing.

Polymeric Component	Antibacterial/Antimicrobial Component/Polymer	Aim and Application	Ref.
PMMA	Ag NPs, OCT	Control the drug release, 3D multilayered porous scaffold diabetic wounds, and repair skin	[76]
PVA/PLA	(PVA-CTX/PLA) and tranexamic acid coagulant (PVA-CTX-TXA/PLA)	Drug release, scaffold can be used to treat burns and chronic and diabetic wound infections	[121]
PCL nanofibers	Photoresponsive nanogel containing Ag NPs	Clinical application as wound dressing activated by light	[114]
PVA and carbon nanotubes	Ag NPs	Antibacterial wound dressing	[125]
PCL, HAP	Ag^+^	Scaffold with appropriate characteristics for wound healing	[111]
PLA and PCL	Ag-chitosan NPs	Tissue regeneration and wound healing	[126]
PLA/chitosan	Ag^+^	High antibacterial activity and a high potential for applications in biomedical fields	[85]
PLA	Chitosan, copper, or silver-doped bioactive glasses	Good in vitro bioactivity of the fibers and possible bone tissue regeneration	[120]
Chitosan, PVA	ZnO NPs	Accelerated wound healing. Nanofibrous mat with antibacterial and antioxidant properties for diabetic wound healing	[116]
PCL, collagen, zein	*Aloe vera*, and ZnO NPs	Biocompatible and non-toxic materials for wound dressing	[127]
PCL, gelatin	ZnO NPs, amoxicillin	treatment of full-thickness wounds	[128]
PMMA	Ag NPs, Octenidin (OCT)	Control the drug release, 3D multilayered porous scaffold diabetic wounds, and repair skin	[129]

Appropriate antibacterial wound dressings play a pivotal role in promoting the healing process of chronic wounds. In this context, incorporating chitosan as an adjunct within the biopolymeric electrospun fiber matrix has shown promising outcomes, exhibiting significant enhancements in both the wound-healing capabilities and antibacterial efficacy of the dressing material (Table 2). In wound care, researchers have sought to enhance the therapeutic effectiveness of wound dressings by incorporating a diverse array of bioactive agents. These agents encompass a wide range of substances, such as antibiotics (e.g., ciprofloxacin, metronidazole, gentamicin, norfloxacin, etc.), metal-based nanoparticles (e.g., Ag and ZnO NPs), plant extracts (including *aloe vera*, curcumin), growth factors, vitamins, and others [118]. By integrating these therapeutic components, the biological activities of nanofibers, commonly utilized in wound dressings, can be significantly elevated. The unique capacity of nanofibers to facilitate drug delivery further underscores their suitability for wound-care applications (Table 3). Ghorbani et al. [128] created electrospun nanofibers for a wound dressing that hybridized collagen, Poly(ε-caprolactone) (PCL), zein, *aloe vera*, and ZnO NPs. These nanofibers had better tensile strength and improved fibroblast cell proliferation and attachment compared to plain nanofibers. This indicates that they are biocompatible and not toxic. Jafari et al. [129] created bilayered nanofibers made of PCL and gelatin infused with amoxicillin and zinc ZnO NPs for treating the bacterial infection of wounds. The nanofibers showed a high swelling degree due to the hydrophilic gelatin. The hybrid nanofibers showed a sustained release of ZnO NPs and amoxicillin, as indicated by in vitro drug release assessments. The antibacterial analysis of the dual drug-loaded hybrid nanofibers using the disk diffusion method in vitro showed a good antibacterial efficacy. The full-thickness rat models underwent in vivo tests, which showed that the fabricated nanofibers help to speed up wound contraction, increase collagen deposition and angiogenesis, and prevent the formation of scars. These tests indicate that the fabricated scaffolds could be a promising treatment option for full-thickness wounds.

**Table 3 pharmaceutics-16-00093-t003:** Antibacterial scaffolds for wound dressing.

Polymeric Component	Antibacterial/Antimicrobial Component	Method of Preparing	Aim and Application	Ref.
Chitosan, bacterial cellulose	Chitooligosaccharide	Composite membranes	Good antioxidant activity and wound-dressing applications	[130]
Collagen (type I)	Tobramycin (Tob)	Film casting	Potential application in corneal repair	[131]
Chitosan	Quaternary ammonium chitosan NPs (TMC NPs)/chitosan	Lyophilization (sponge composite)	Promising dressing material for chronic wounds	[37]
Curcumin-β-cycyclodextrin inclusion complex (CMx) and cellulose	Chitosan	Freeze drying (sponge composite)	Wound-dressing materials for the treatment of wounds (especially chronic wounds)	[132]
Chitosan	Hydroxybutyl chitosan	Freeze drying (sponge composite)	Excellent antibacterial activity of composite sponge to be applied for wound dressings	[133]
Chitosan	Chitosan	3D printing	Improving the quality of the restored tissue concerning both commercial patches and spontaneous healing	[134]
PCL and silk sericin	Chitosan/sodium alginate hydrogel	Electrospin ning and 3D bioprinting	Promoting the healing process and skin tissue engineering	[135]
Chitosan-PEG	Cu NPs	Hydrogel	Sustained drug release with excellent keratinocyte cell response and anti-infection wound dressing	[136]

### 4.2. Biofunctionalized Antibacterial Nanofibers for Wound Dressings

Biofunctionalized antibacterial nanofibers represent a specific type of wound-dressing material characterized by the surface functionalization of biopolymeric nanofibers with amino acids and AMPs [137]. Chitosan and silk fibroin (SF) are the two key biopolymers used in biofunctionalized nanomaterials due to their ability to facilitate the attachment of various antimicrobial agents through multiple functional groups. Extensive research has been conducted on AMPs that are immobilized on the surface of nanofibers [138,139,140,141].

#### 4.2.1. Silk Fibroin (SF)

Recent research indicates that SF exhibits notable biocompatibility, favorable mechanical characteristics, and promising physiological properties, making it a viable option for applications in the clinical industry, as well as in the medical sectors, especially in wound dressings [142].

Numerous antibacterial biohybrid nanofibrous wound dressings are created by leveraging the surface functionality of SF. The various functional groups, such as phenol, carboxyl, hydroxyl, and amines, are loaded into SF nanofibers [143]. It has been observed that SF biohybrid nanofibers effectively inhibit bacterial growth [144]. Higher amounts of immobilized factors result in greater antibacterial activity. However, it has been discovered that the bacterium *S. aureus* can compromise the efficiency of AMPs by altering the negative surface charge, modifying the membrane fluidity, or employing efflux pumps to keep the AMPs at bay [145]. Another investigation focused on the impact of fibroin morphology on the release of silver ions and its concurrent antibacterial activity against *S. aureus*, *S. epidermidis*, and *P. aeruginosa* [92]. Further, researchers employed endocrine-disrupting chemicals/N-hydroxysuccinimide and thiol–maleimide click chemistry techniques to immobilize an antimicrobial peptide motif (Cys-KR12) derived from the human cathelicidin peptide (LL37) onto electrospun SF nanofiber membranes. The resulting nanofiber membrane exhibited antimicrobial activity against four pathogenic bacterial strains, namely, *S. aureus*, *S. epidermidis*, *E. coli*, and *P. aeruginosa* [108]. Hadisi et al. [146] fabricated a nanocomposite containing gentamicin (GEN), hardystonite (HT), and SF. The results obtained from measuring the antibacterial inhibition zone indicated a significant augmentation in the antibacterial properties of the scaffolds against *E. coli* and *S. aureus* bacteria when incorporating GEN at concentrations ranging from 3 to 6 wt%.

#### 4.2.2. Chitosan

Chitosan is a biopolymer known for its biocompatibility and biodegradability, and it exhibits notable antimicrobial properties against various microorganisms, encompassing bacteria, algae, viruses, and fungi [147]. Cai et al. [148] assessed the antimicrobial effectiveness of composite nanofibers containing chitosan against two types of bacteria, namely the Gram-negative bacterium *E. coli* and the Gram-positive bacterium *S. aureus*, using the turbidity measurement method. Ultimately, the tested nanofibrous mats could certainly be used as a wound dressing. Other researchers created an electrospun mat using chitosan, known for its biocompatibility. Incorporated into the structure, the photosensitizer endowed the material with light-induced and spatially restricted antimicrobial properties, as evidenced by its effectiveness against *S. aureus* [149].

To create biocompatible antimicrobial nanofiber wound dressings, two natural extracts, *Cleome droserifolia* (CE) and *Allium sativum* aqueous extract (AE), were incorporated into honey, PVA, and hydroxypropyl chitosan (HPCS). The results were compared with those of the commercial dressing Aquacel Ag. The study revealed that the HPCS–AE and HPCS–AE/CE NF mats completely inhibited *S. aureus*. However, the HPCS–AE/CE exhibited mild antibacterial activity [150].

Moreover, chitosan has found applications as a delivery system. For example, Moursa et al. [151] utilized chitosan to deliver neurotensin, a neuropeptide known for its inflammatory modulatory effects, to treat diabetic foot ulcers (DFUs). Wound dressings incorporating a chitosan–collagen complex have also been investigated for their efficacy in treating thermal skin burns. These dressings were found to promote an accelerated healing process by facilitating the formation of granulation and fibrous tissue [152]. Table 4 shows natural and synthetic material systems used for dressings.

## 5. Hydrogels with Nanofibers and Functional Thermoresponsive Hydrogels as Wound Dressings

### 5.1. Hydrogels with Nanofibers

Hydrogels represent hydrophilic materials composed of natural and synthetic polymers, such as certain gelatin, alginate, poly(methacrylates), and poly(vinyl pyrrolidone). These materials are insoluble and have a high water content, typically from 70% to 90%. This moisture in hydrogels fosters an environment conducive to granulation tissues and the epithelium, promoting wound healing. Hydrogels’ soft and elastic nature allows easy application and removal after the wound has healed, ensuring minimal damage [162]. However, owing to their substantial water content, hydrogels exhibit a restricted absorptive capacity [59]. Many studies collectively represent significant progress in the development of novel treatments for wounds, emphasizing the potential of hydrogel-nanofiber-loaded technologies in advancing wound care and biomedical applications [2,119].

For instance, Ren et al. recently conducted a study in which they incorporated electrospun nanofibers made of PVA/sodium alginate that contained hesperidin and Ag nanoparticles (Ag-Hes NPs) into a hydrogel (Ag-Hes@H). The fiber–hydrogel combination has demonstrated encouraging outcomes with regard to the healing of infected wounds in animal experiments. Moreover, it functions as an antibacterial and anti-inflammatory agent and promotes the growth and movement of skin cells (Figure 6) [163].

In another study, an investigation was conducted into a nanocomposite made of antibiotic-loaded hydrogel and electrospun fibers grafted with poly(gallic acid), which exhibited antimicrobial and antioxidant properties. The nanocomposite was found to have no cytotoxicity, as evidenced by the absence of hemolytic activity and the viability of epithelial cells [164]. In a study conducted by Li et al. [165], a novel approach was employed to enhance the healing of diabetic wounds. This approach involved the fabrication of scaffolds using poly(d,l-lactic acid) (PDLLA) nanofibers and gelatin methacryloyl (GelMa) hydrogels with a three-dimensional multilayer patterned structure. The scaffold adopted a nanofiber/hydrogel core–shell configuration, which exhibited exceptional exudate-absorption capabilities, creating a moist wound environment. It significantly stimulated the development of a three-dimensional capillary network. Consequently, this approach accelerated diabetic wound healing, highlighting the potential of such scaffolds for chronic wound recovery (Figure 7). In innovative studies, researchers have explored various treatments for chronic ulcerative wounds, focusing on hydrogel-based nanofiber materials. Chen and co-authors [131] developed layer-by-layer self-assembled peptide hydrogel nanofibers by modifying N-acetaminophen glucose. This novel approach demonstrated potent antibacterial properties and significant contributions to angiogenesis and wound healing. Another study, conducted by Zhong et al. [166], created a dynamic reversible borate ester bond using dopamine-grafted oxidized carboxymethyl cellulose and cellulose nanofibers (CNFs). The result was a hydrogel dressing with remarkable self-healing properties. Notably, the dressing also demonstrated the ability to degrade over time, making it an up-and-coming solution for medical applications. Additionally, Zhong et al. [167] developed a unique hydrogel dressing by utilizing a dynamic covalent bond between boric acid and a catechol group, enabling the loading of epigallocatechin-3-gallate into quaternized chitosan. This integration allows for rapid self-healing properties. Furthermore, hydrogels can reduce the temperature of cutaneous wounds, offering a soothing and cooling effect [168].

In a recent investigation, it was demonstrated that the incorporation of silicate nanosheets (Laponite) into gelatin methacryloyl (GelMA) hydrogels resulted in noteworthy enhancements in both mechanical attributes and adhesion. Furthermore, these modified hydrogels exhibited a prolonged release of epidermal growth factor (EGF) and showed the capacity to halt bleeding, facilitating a comprehensive process of skin regeneration [169]. Recently, Xu unveiled a novel composite hydrogel system. This hydrogel framework is fortified through the use of a colloidal blend involving carbon nanotubes (CNTs) and GelMA. Subsequently, it experiences in situ polymerization while also integrating antimicrobial peptides, leading to substantial enhancements in both the electrical conductivity and mechanical characteristics of the hydrogel. Additionally, it was observed that the GelMA component within this hydrogel construct is conducive to promoting cell adhesion and proliferation [170].

### 5.2. Thermoresponsive Hydrogels with Antibacterial Properties

Supramolecular hydrogels that use non-covalent interactions, like hydrophobic or ionic interactions, to achieve gelation usually indicate a high sensitivity to various external stimuli like the temperature or pH. In the case of temperature-sensitive gelation, they are called thermoresponsive hydrogels (Figure 8). These hydrogels have polymers that contain both hydrophilic and hydrophobic components, making them amphiphilic. When exposed to elevated temperatures, these thermosensitive hydrogels, also known as thermogels, undergo a distinct phase-transition behavior. Unlike conventional melt transitions, they change from a liquid to a gel state and can revert to a liquid state at lower temperatures. The gelation process is self-initiated, meaning that it does not require external aid or enzymatic catalysts. This makes it a gentle phase-conversion technique [171]. Contemporary wound dressings, such as thermoresponsive hydrogels, are typically prepared beforehand. This method can result in uncovered areas of the wound and decreased effectiveness regarding therapy and antibacterial properties, particularly for wounds of uneven shape. Injectable hydrogels, however, can be customized to fit the shape of irregular wounds, making them more efficient. Injectable hydrogels are made by in situ gelation, mainly achieved via chemical crosslinkers such as enzymatic catalysis, photo-initiated crosslinking, and Schiff–base reactions [171,172,173,174]. In a recent study, researchers utilized thermoresponsive poloxamer (P407)–PVA hydrogels to deliver mupirocin nanoparticles containing either gelatin or poly(acrylic acid) for wound healing. The study’s findings indicate that this method can effectively treat wound infections. This technique could potentially be applied to pharmaceutical products to aid in their removal from wounds and harness their antimicrobial properties [173]. A new thermoresponsive hydrogel has been developed by researchers. It is a mixture of galactose-modified xyloglucan and hydroxybutyl chitosan that is intended to act as a barrier to prevent re-adhesion after adhesiolysis. The hydrogel can be injected and will solidify at body temperature, making it extremely practical. This composite hydrogel has been demonstrated to be effective in preventing adhesion, aiding wound healing and reducing scarring, making it a promising injectable anti-adhesion solution for various clinical applications [171]. Poloxamer hydrogels exhibit thermoresponsive behavior with fluidity upon cooling and a gel-like consistency upon heating [172,174]. Niyompanich et al. [172] conducted a study with Poloxamer hydrogels that displayed the rapid gelation time of approximately 95 s. Biocompatibility with L929 cells persisted even after seven days of cell culturing. The gentamicin-loaded poloxamer hydrogels exhibited immediate bacterial growth inhibition, with evidence of synergistic effects. Notably, these hydrogels showed larger inhibition zones compared to equivalent gentamicin solution concentrations. Therefore, the hydrogel composed of 20 wt% poloxamer (407) and 3 wt% poloxamer (188) dissolved in water holds promise as a potential drug carrier for cavity wounds [172].

Chemically crosslinked injectable hydrogels may be less suitable for biomedical applications due to the presence of potentially toxic small molecule additives used in the crosslinking process [38,172]. Various thermoresponsive polymers can undergo sol–gel transition through hydrophobic interactions, forming injectable hydrogels after the phase-transition temperature is reached. Notably, poly(N-isopropylacrylamide) (PNIPAM), with a lower critical solution temperature just below human body temperature (37 °C), has garnered significant interest for its biomedical potential [171,175,176]. In a mouse model, Mahdieh et al. [175] investigated controlled delivery using core–shell electrospun fibers and thermoresponsive PNIPAM hydrogel particles. The fibers contained Ag NPs of various sizes and ZnO NPs aimed at improving the pore structure and precise Ag NP release. A 27-day in vivo mouse implantation study observed consistent and regulated Ag NP release. Hyperspectral imaging revealed distinctive patterns in released Ag NPs in male and female mice. Male mice with ZnO NP-loaded fiber implants exhibited enhanced hair regrowth and wound healing, countering in vitro cytotoxicity findings. These results signify the potential of these novel fiber meshes for sustained drug release and compatibility while also suggesting the feasibility of gender-specific drug delivery systems (Figure 8) [175].

A hydrogel that can be triggered by skin temperature is being developed to offer versatile and antibacterial wound protection. A novel dual-thermoresponsive hydrogel incorporating PNIPAM and methacrylated κ-carrageenan was produced by Feng et al. and exhibited shape adaptability at physiological temperatures. It synergistically integrates near-infrared-responsive (NIR-responsive) polypyrrole–polydopamine nanoparticles (PPy-PDA NPs) and Zn^2+^ derived zeolitic imidazolate framework (ZIF-8), enabling localized NIR heat generation and controlled Zn^2+^ release. Studies have shown that utilizing NIR heating in this hydrogel can accelerate the release of Zn^2+^, which enhances its efficacy in combating severe infections. This hydrogel holds great promise as a wound dressing, as it has the ability to adapt to the contours of the wound, facilitate tissue regrowth, and provide sustained antibacterial defenses through the synergistic interplay of photothermal and chemical mechanisms [38]. A temperature-manipulable material, responsive to cold water, shows potential for the pain-free removal of wound dressings. Radhakumary et al. designed a novel formula with thiolated chitosan and PNIPAM containing ciprofloxacin, an extensive antibacterial drug. This composite is a smart, drug-eluting mucoadhesive gel that combines natural and synthetic attributes. The cytocompatible material enables sustained ciprofloxacin release, indicating prolonged wound protection. The thermoresponsive thin film swells with cold water, facilitating gentle removal without skin trauma. With suitable mechanical properties for wound care, the film exhibits cytocompatibility and controlled antibacterial release for over 48 h [177]. Nitric oxide (NO) plays a role in various physiological processes, such as vasodilation, wound-healing, and antibacterial activities. Recent studies found NO was released from the thermoresponsive hydrogels composed of S-nitrosoglutathione (GSNO)/Pluronic F-127 (PL) combined with natural polysaccharides polymers (chitosan and alginate). The results showed a sustained release of NO from GSNO hydrogels. Furthermore, antibacterial activity, cytotoxicity, wound healing, and good mechanical stability are the most important characteristics of these hydrogels [178,179]. The hydrogel displayed a strong ability to kill bacteria, including methicillin-resistant *S. aureus* and multidrug-resistant *P. aeruginosa*. When used to treat wounds infected with *P. aeruginosa*, it also promotes faster healing and reduces the number of bacteria present in the wound. The GSNO-PL/Alginate thermoresponsive hydrogel can be potentially used to treat infected wounds [178]. The GSNO-PL/chitosan thermoresponsive hydrogel could be used for topical NO delivery as a simple and cost-effective method [179].

Finally, to enhance wound healing and tissue regeneration, it could be advantageous to merge polymers of thermoresponsive and biodegradable characteristics with antibacterial bioactive materials containing diverse drug types. This combination can produce encouraging outcomes.

## 6. Conclusions

Recent advancements in wound-dressing materials have shown promise in addressing the challenges associated with ideal wound care. Notably, electrospun and hydrogel materials, potentially incorporating nano-additives, emerge as key players in innovative treatment approaches. Electrospun antimicrobial drug-delivery systems, characterized by tunable porosity and a large specific surface area, demonstrate significant potential in biomedicine, particularly for wound dressings. These fibers allow for surface engineering with functional groups and drug immobilization. Similarly, thermoresponsive hydrogels represent a dynamic shift in wound care, offering a versatile approach to wound management and antibacterial treatment. The integration of responsive polymers and bioactive agents holds considerable promise for advancing wound-healing strategies, leading to patient-centric wound dressings.

In conclusion, further research involving a combination of different materials is essential. The effectiveness of delivery systems, along with advancements in scaffold formulations based on biomaterials, explains the growing inclination to explore their potential in wound-healing preparations. As challenges, such as clinical-level issues, are addressed, electrospun and hydrogel materials may experience a significant breakthrough in biomedical applications in the near future.

## Figures and Tables

**Figure 1 pharmaceutics-16-00093-f001:**
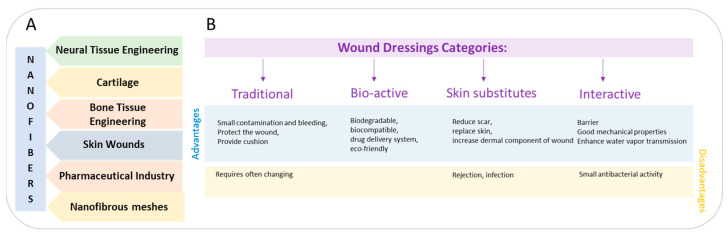
(**A**) Illustration of nanofiber applications in different fields of biomedicine, and (**B**) classification of wound dressings.

**Figure 2 pharmaceutics-16-00093-f002:**
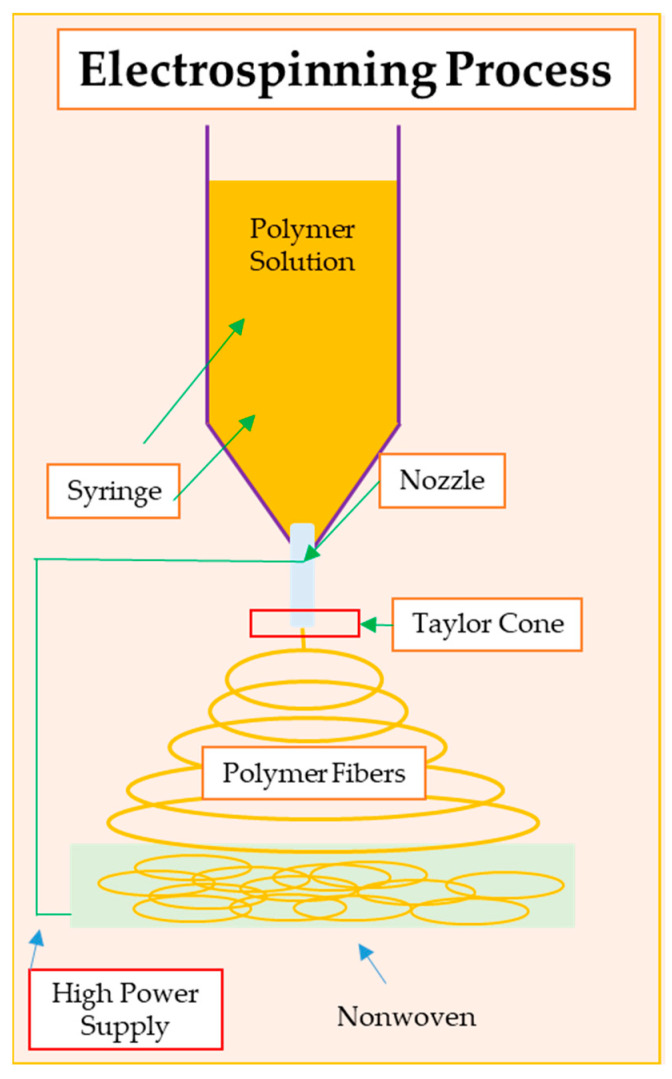
Scheme of the electrospinning process.

**Figure 3 pharmaceutics-16-00093-f003:**
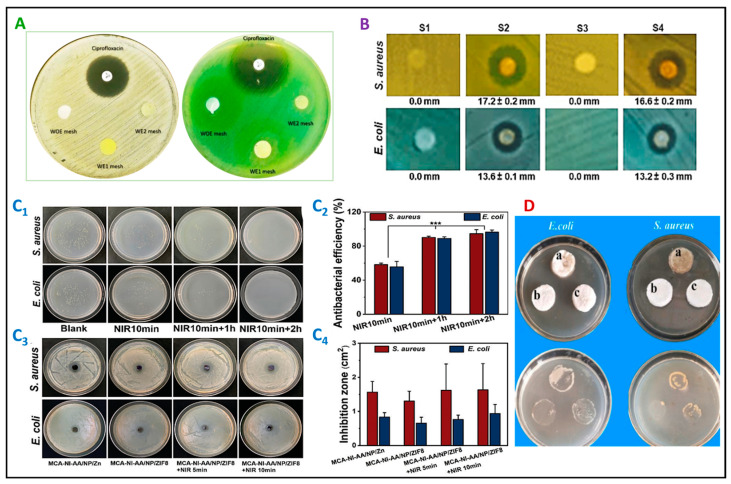
Different structures of scaffolds with antibacterial properties. (**A**) Versatile Gelatin/Chitosan electrospun wound dressing exhibiting antimicrobial effects against *P. aeruginosa* and *S. aureus* through the use of ciprofloxacin (CIP), with the permission of Licensee MDPI Copyright 2021 [84]. (**B**) Dual-drug delivery via core–shell PVA/PCL electrospun nanofibers loaded with Ag-chitosan nanoparticles and phenytoin for tissue regeneration and wound healing, with the permission of Elsevier Copyright 2021 [85]. (**C_1_**–**C_4_**) Dual-thermoresponsive hydrogel, comprising poly(N-isopropyl acrylamide) (PNIPAM) and methacrylated κ—carrageenan (MA-K-CA) with near-infrared (NIR)-responsive polypyrrole–polydopamine nanoparticles (PPy-PDA NPs), Zn^2+^ and ZIF-8, designed for bacterial elimination during wound healing. (**C_1_**) Visual documentation of colonies, and (**C_2_**) Antibacterial performance of the MCA-NI-AA/NP/ZIF8 hydrogel against *S. aureus* and *E. coli* (*** *p* < 0.001)). All values are expressed as mean ± SD, n = 3. (**C_3_**) Captured images of the inhibition zone, and (**C_4_**) statistical data regarding the MCA-NI-AA/NP/Zn inhibition zone area and the MCA-NI-AA/NP/ZIF8 hydrogel against *S. aureus* and *E. coli*. This hydrogel represents a promising wound dressing with adaptable coverage and photothermal–chemical antibacterial capability, ensuring effective bactericidal action and prolonged release of antibacterial agents, with the permission of Wiley-VCH GmbH Copyright 2022 [38]. (**D**) A novel quaternary ammonium chitosan nanoparticle (TMC NP)/chitosan composite sponge with asymmetric wettability surfaces is a promising dressing material for chronic wounds. The bacterial infiltration activity of chitosan (a), modified TMC NP/chitosan (b), and modified chitosan (c) sponges against *E. coli* and *S. aureus*, with the permission of Elsevier Copyright 2019 [37].

**Figure 4 pharmaceutics-16-00093-f004:**
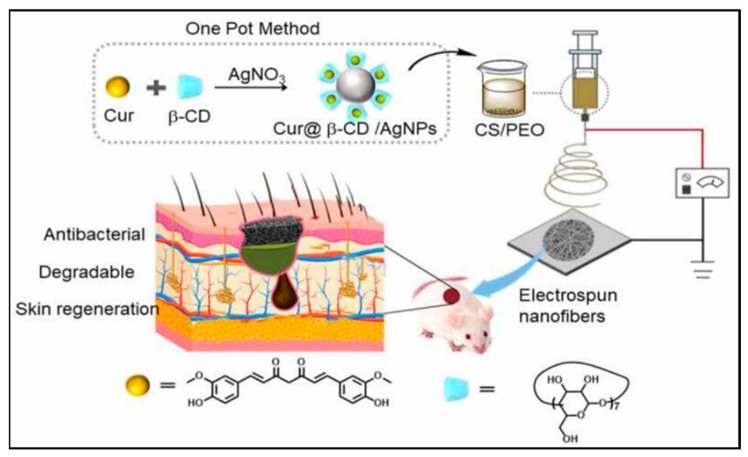
Schematic illustrating the impact of a wound dressing comprising a blend of silver@curcumin NPs and electrospun chitosan nanofibers on the process of wound healing. Electrospun nanofiber scaffolds containing chitosan and curcumin enhance tissue regeneration by promoting angiogenesis and tissue proliferation while reducing scar formation. This study provides an efficient method for producing advanced nanofibrous scaffolds with strong antibacterial and wound-healing properties. The figure is reprinted with the permission of Taylor & Francis Group Copyright 2022 [96].

**Figure 5 pharmaceutics-16-00093-f005:**
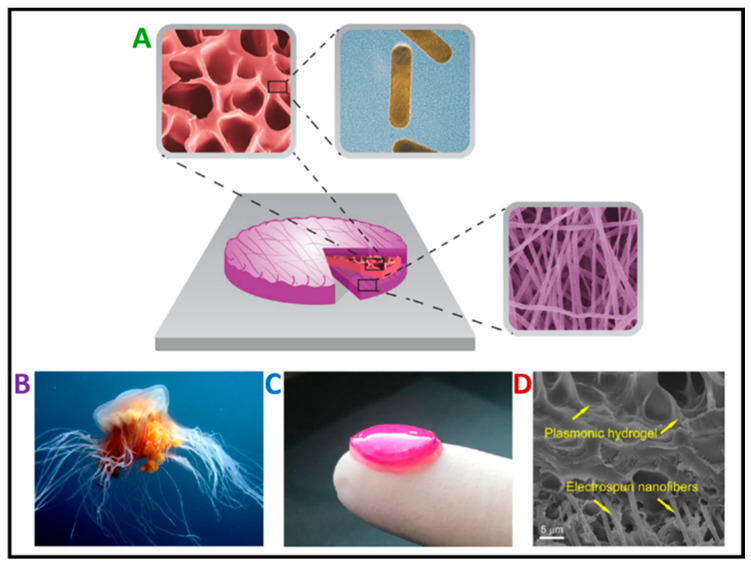
Smart nanostructured pillow for diverse biomedical applications powered by near-infrared light. (**A**) Schematic of the hierarchical platform: electrospun fiber and hydrogel with gold nanorods (Au NRs). (**B**) Photograph of Phacellophora camtschatica jellyfish near Gibraltar, courtesy of Prof. Stefano Piraino (University of Salento, Italy). (**C**) Image of the nanostructured pillow with rhodamine B (RhB). (**D**) Cross-section showing the hydrogel within the electrospun fiber, creating a biomimetic design for biomedical applications, with the permission of ACS Copyright 2020 [119].

**Figure 6 pharmaceutics-16-00093-f006:**
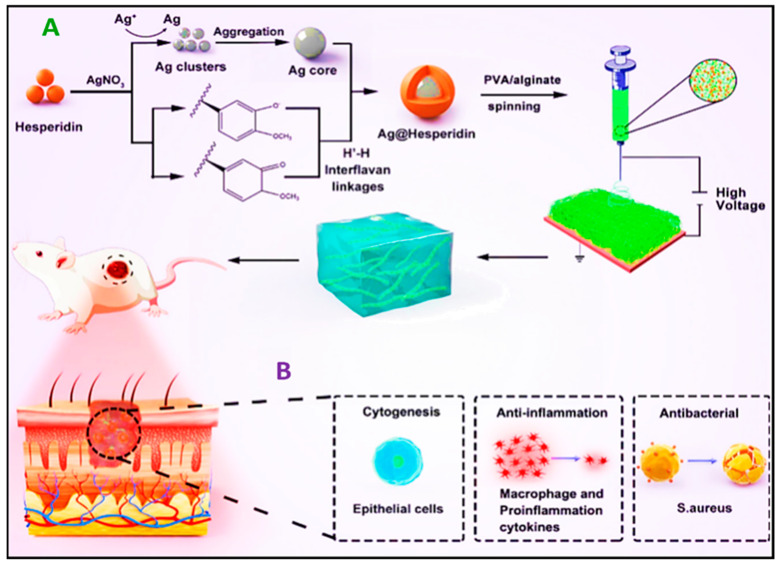
Schematic demonstrating the preparation of electrospun nanofibers containing Ag@hesperidin (Ag@Hes) core–shell nanoparticles fabricated to promote the healing of infected wounds by providing antibacterial and anti-inflammatory properties. Following this, (**A**) illustrates the sequential procedure for producing Ag-Hes NPs, electrospinning nanofibers loaded with Ag-Hes NPs, and Ag-Hes@H. Subsequently, (**B**) outlines potential mechanisms by which Ag-Hes@H enhances the recovery of infected wounds during animal experiments. The figure is reprinted with the permission of Oxford University Press Copyright 2022 [163].

**Figure 7 pharmaceutics-16-00093-f007:**
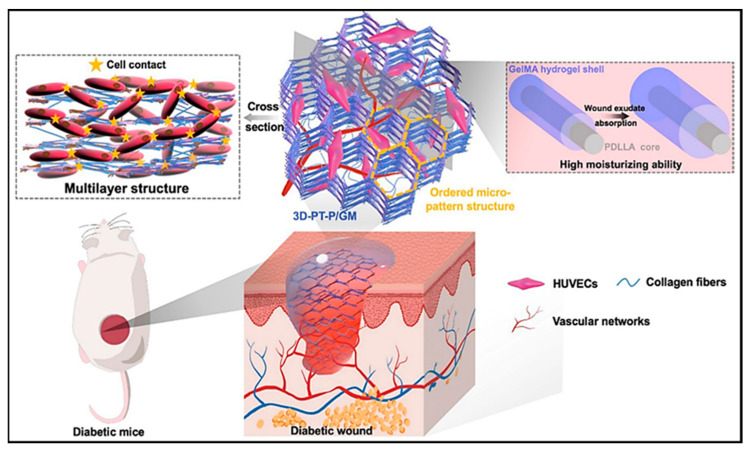
Schematic illustrating the core–shell hydrogel/nanofiber (with GelMA hydrogel as the shell and PDLLA as the core) composite scaffold with the 3D patterned structure for diabetic wound healing, reprinted with the permission of BMC, Springer Nature Copyright 2022 [165].

**Figure 8 pharmaceutics-16-00093-f008:**
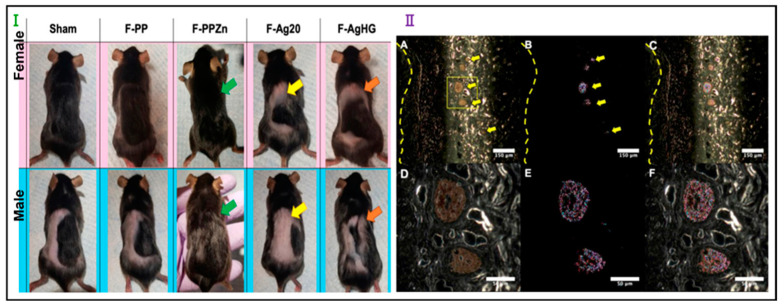
Schematic representation of the core–shell electrospun fibers (containing ZnO NPs) loaded with silver nanoparticles (Ag NPs) where the delivery rate was controlled by different sizes of Ag NPs and thermoresponsive PNIPAM hydrogel particles. (**Ⅰ**) The skin appearance 27 days post surgery shows the tolerance of fiber meshes with varying NP contents. Female mice exhibit greater hair regrowth than males, with the most significant regrowth observed in the F-PPZn group (green arrows). The presence of Ag NP in the F-Ag20 and F-AgHG groups (yellow and orange arrows) reduces hair regrowth. (**Ⅱ**) Hyperspectral imaging of male mouse skin 27 days post implantation of F-AgHG fiber mesh demonstrates continuous Ag NP release into the dermis. Photomicrographs (**A**–**C**) at 100× magnification and (**D**–**F**) at 400× magnification reveal dispersed Ag NPs in the dermis. Yellow dashed lines denote the tissue–fiber-mesh interface. Scale bars: 150 μm (**A**–**C**) and 50 μm (**D**–**F**), reprinted with the permission of Elsevier Copyright 2021 [175].

**Table 1 pharmaceutics-16-00093-t001:** Available wound dressings and their advantages and disadvantages.

Wound Dressing Type	Wound Type	Advantages	Disadvantages	Examples	Ref.
Traditional wound dressings					
Tulle	Shallow wounds	Does not adhere to the wound	Insufficient, often additional dressing required	Paratulle, Jelonet, Bactigras	[41]
Gauze	Minor clean wounds	Cheap, used as cover	Frequent changing, may adhere to the wound	Xeroform	[42]
Interactive and bioactive wound dressings					
Hydrogels	Dry wounds, wounds with low-to-medium exudate, necrotic wounds, burn wounds	Maintains moisture, allows vapor and oxygen exchange, does not react with tissue	Infections of the skin, mechanically weak	Intrasite Hydrosorb, Transigel, Curafil, FlexiGel, Aquaform Vigilon, Curasol	[43]
Semi-permeable films	Shallow wounds	Flexible, conformable, allow gas exchange	Minimal adhesion, skin maceration, dryness	Opsite, DuoDERM, Bioclusive, Mefilm, Transeal, Tegaderm, Omniderm	[44]
Semi-permeable foams	Moderate wounds, heavy wounds	Highly absorbent	Dryness	Lyofoam, Mepilex, Allevyn Curafoam, Polymem,	[45]
Hydrocolloids	Minor burns, light wounds, moderate exuding wounds	Absorption and debridement of wound exudates, permeable to water vapor, fluid exchange	Not intended to be used for heavy wounds	Granuflex, Tegasorb, DuoDERM, Replicare, NuDerm, Tegasorb	[46]
Hydrofibres	Burns, medium wounds, heavy wounds	Highly absorbent	Secondary dressing	Aquacel	[47]

**Table 4 pharmaceutics-16-00093-t004:** Examples of electrospun materials with process parameters dedicated to wound-dressing applications.

Material	Method of Synthesis	Process Parametres	Properties Investigated	Ref.
Chitosan/PEO/semelil	Electrospinning	V—10–15 kV FR—0.45–0.60 mL/h T-C-D—10–15 cm.	Semelil release	[153]
PCL/PVA/curcumin	Forcespinning		Biocompatibility, anti-bacterial property, absorption	[154]
Chitosan/PVA/Nepeta dschuparensis/honey	Electrospinning	V—17 kV FR—0.5 mL/h	In vivo properties, biocompatibility, biodegrability	[155]
PCL–silk fibroin/silk fibroin–hyaluric acid–Thymol	Electrospinning	V—30 kV FR—2.3 mL/h, 12 cm	Biocompatibility, biodegrability, antibacterial property	[156]
Polycaprolactone/tyrosol/Thymol	Electrospinning	V—28 kV FR—2.3 mL/h, T-C-D—12 cm	Antibacterial property	[154]
Cellulose acetate/β-cyclodextrin/Thymol	Electrospinning	V—+6.62 to +10.22 kV FR—1.0 mL/h	Antibacterial activity, drug release	[157]
PCL/PVA/Eugenol/Chitosan	Electrospinning	V—75.0 kV FR—13 cm	Release Eugenol, biocompatible, non-toxic, antibacterial properties	[158]
Silk fibroin/fenugreek/collagen	Electrospinning	V—25 kV FR—0.5 mL/h, T-C-D—10 cm	Biocompatible, wound healing, antioxidant property	[159]
PCL/lawsone/gelatin	Electrospinning	15 kV 1.19 mL h^−1^ 14 cm	Wound healing, antibacterial properties, biocompatible, healing	[160]
Polyurethane/silver/cellulose acetate/graphene oxide/curcumin	Electrospinning	17 kV 0.4 mL/h. 15 cm	Biocompatible, promote wound healing, antibacterial property	[161]

V—voltage; FR—feed rate of the polymer during the process; T-C-D—tip-to-collector distance.

## Data Availability

Not applicable.

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
