# Peer review of "Electrospun Antimicrobial Drug Delivery Systems and Hydrogels Used for Wound Dressings"

_pharmaceutics, 2024, doi:10.3390/pharmaceutics16010093_

Round 1

Reviewer 1 Report

Comments and Suggestions for Authors

This manuscript reviewed the progress in wound dressing materials, especially electrospun nanofiber materials for antimicrobial drug delivery and thermoresponsive hydrogels. The main idea of the paper is unclear and not very relevant. It is suggested focus on one subject, such as antimicrobial properties for wound healing electrospun materials. There are some other concerns as follow.

1.     In section 3, the introduction for the electrospinning wound dressing is not adequate. For example, usually what kind of materials are used for the construction of wound dressing, how the fabrication parameters affect the wound dressing properties, and how these materials fulfill would repair function?

2.     In section 4, the incorporation of antibacterial agents is common for wound dressing materials, the authors should focus on the advantages of electrospinning for drug release or wound healing and their mechanism.

3.     Fig.1A repeat for the “Bone Tissue Engineering”.

4.     Fig.2 is not suitable as the introduction of electrospinning in this part.

Reviewer 2 Report

Comments and Suggestions for Authors

The authors of this manuscript offer a comprehensive review focusing on electrospun dressings that integrate antibacterial elements. Among the array of available technologies, electrospun fibers possess distinct attributes that make them promising materials for new dressings. Electrospinning represents a straightforward technique for generating drug-containing fibers of varied diameters. Moreover, it enables the integration of diverse types of drugs, facilitating controlled release of biological agents. Consequently, this subject holds significant importance and relevance in the current context.

I think authors should incorporating a summary table detailing natural and synthetic polymers utilized in electrospun fibers for wound healing, along with their respective electrospinning parameters and the solvents employed during the electrospinning process, would significantly enhance the manuscript's comprehensiveness. This table would offer a quick reference guide, consolidating essential information on material compositions, manufacturing conditions, and key components utilized in various electrospun wound dressings, thereby enriching the overall scope and usefulness of the manuscript.

While electrospinning possesses many advantages in drug delivery and tissue engineering, which are beneficial for wound healing, concerns over the use of harsh chemicals/solvents (cytotoxicity) may limit its use in pharmaceutical applications for dressing materials. I would like to know the authors comments on this

Reviewer 3 Report

Comments and Suggestions for Authors

Reviewer’s comments:

The manuscript entitled ‘Electrospun antimicrobial drug delivery systems and thermoresponsive hydrogels used for wound dressings – a systematic review’ has been peer-reviewed. The authors have explained recent wound dressing materials, their formation, application, and functionalization and highlighted Recent advancements in wound care have highlighted the potential of thermoresponsive hydrogels as dynamic and antibacterial wound dressings. We have provided the following comments to improve the manuscript.

Minor concern

1) Wounds and chronic wounds can be caused by bacterial infections and are leading to discomfort for patients, and require healthcare attention.

Wounds and chronic wounds. Are they separate specific terms?

2) Figure 1. Engance. Is it ‘enhanced’?

3) The flow of manuscript content (including title) is confusing. What do the authors want to highlight and specify? There is confusion with phrases like ‘Antimicrobial electrospun nanofiber’, ‘antimicrobial thermoresponsive hydrogel’, ‘hydrogel containing electrospun nanofiber’, and ‘functional thermoresponsive hydrogel’.

Comments on the Quality of English Language

Minor editing of the English language is required.

Reviewer 4 Report

Comments and Suggestions for Authors

The comments for the authors are the following:

1. The purpose of the review paper does not coincide with the data that are exemplified. Please highlight the purpose of this review, it is not clear if you are addressing topics only with hydrogels used for wound dressings or other types of dressings. In the abstract, you only mention hydrogels.

2. The title of the manuscript is not clear; authors should modify the title.

3. The abstract must be rewritten. Add the novelty to the abstract.

4. In Figure 1A, you don't mention nanofibrous mesh anywhere

5. Table 2, chitosan is not an antibacterial/microbial compound, it is a polymer. Please change the text.

6. Table 3 is not mentioned in the text, please review the data regarding the preparation method. Collagen film is not the preparation method

7. Please pay attention to the citations, there are many phrases without references.

8. The conclusion is too lengthy. It should be more concise.

Comments on the Quality of English Language

The English language needs to be a little polished

Round 2

Reviewer 2 Report

Comments and Suggestions for Authors

The revised manuscript is ok for me

Reviewer 4 Report

Comments and Suggestions for Authors

The authors have improved the paper, but there are still some minor comments:

1. In table 2, the word polymer should not be added there, you already present the polymer composition on the left. The polymer composition is Chitosan/PLA and the antimicrobial component is Ag, as highlighted in reference [85].

2. Please change the name of method of formation in table 4, it does not seem appropriate, maybe the method of synthesis.

3. Pay attention to the number of figures, starting with Figure 6 and make the corrections in the text as well.